# Is Fetal Hydrops in Turner Syndrome a Risk Factor for the Development of Maternal Mirror Syndrome?

**DOI:** 10.3390/jcm11154588

**Published:** 2022-08-05

**Authors:** Ivonne Alexandra Bedei, Alexander Graf, Karl-Philipp Gloning, Matthias Meyer-Wittkopf, Daria Willner, Martin Krapp, Sabine Hentze, Alexander Scharf, Jan Degenhardt, Kai-Sven Heling, Peter Kozlowski, Kathrin Trautmann, Kai Jahns, Anne Geipel, Ismail Tekesin, Michael Elsässer, Lucas Wilhelm, Ingo Gottschalk, Jan-Erik Baumüller, Cahit Birdir, Felix Zöllner, Aline Wolter, Johanna Schenk, Tascha Gehrke, Corinna Keil, Jimmy Espinosa, Roland Axt-Fliedner

**Affiliations:** 1Department of Prenatal Diagnosis and Fetal Therapy, Justus-Liebig University, 35392 Giessen, Germany; 2Prenatal Medicine and Genetics München, 80639 Munich, Germany; 3Center for Prenatal Diagnosis, Mathias-Spital Rheine, 48431 Rheine, Germany; 4Center for Prenatal Medicine and Human Genetics, 20357 Hamburg, Germany; 5Center for Prenatal Medicine on Elbe, 20457 Hamburg, Germany; 6Center for Human Genetics, Cytogenetic Laboratory Heidelberg, 69120 Heidelberg, Germany; 7Center for Prenatal Medicine, 55116 Mainz, Germany; 8Praenatal Plus, Center for Prenatal Medicine and Genetics, 50672 Cologne, Germany; 9Center of Prenatal Diagnosis and Human Genetics, 10117 Berlin, Germany; 10Praenatal.de, Prenatal Medicine and Genetics Düsseldorf, 40210 Düsseldorf, Germany; 11Center for Prenatal Medicine “am Salzhaus”, 60311 Frankfurt, Germany; 12Department of Internal Medicine, Johannes Gutenberg University, 55131 Mainz, Germany; 13Obstetrics and Prenatal Medicine, University Hospital Bonn, 53127 Bonn, Germany; 14Prenatal Medicine Stuttgart, 70173 Stuttgart, Germany; 15Department of Gynecology and Obstetrics, Heidelberg University Hospital, 69120 Heidelberg, Germany; 16Westend Ultrasound, Center for Prenatal Diagnosis and Fetal Echocardiography, 60325 Frankfurt, Germany; 17Division of Prenatal Medicine, Department of Obstetrics and Gynecology, University of Cologne, 50931 Cologne, Germany; 18Gynaecologikum, Frankfurt, 60389 Frankfurt, Germany; 19Department of Obstetrics and Gynecology, University Hospital Carl Gustav Carus Dresden, 01307 Dresden, Germany; 20Department of Prenatal Medicine and Fetal Therapy, Philipps University, 35041 Marburg, Germany; 21Baylor College of Medicine Department of Obstetrics and Gynecology, Division of Fetal Therapy and Surgery and Texas Children’s Hospital Fetal Center, Houston, TX 77030, USA

**Keywords:** Mirror syndrome, fetal hydrops, Turner syndrome, monosomy X, placental hydrops, placentomegaly

## Abstract

Mirror syndrome is a rare and serious maternal condition associated with immune and non-immune fetal hydrops after 16 weeks of gestational age. Subjacent conditions associated with fetal hydrops may carry different risks for Mirror syndrome. Fetuses with Turner syndrome are frequently found to be hydropic on ultrasound. We designed a retrospective multicenter study to evaluate the risk for Mirror syndrome among pregnancies complicated with Turner syndrome and fetal hydrops. Data were extracted from a questionnaire sent to specialists in maternal fetal medicine in Germany. Out of 758 cases, 138 fulfilled our inclusion criteria and were included in the analysis. Of the included 138, 66 presented with persisting hydrops at or after 16 weeks. The frequency of placental hydrops/placentomegaly was rather low (8.1%). Of note, no Mirror syndrome was observed in our study cohort. We propose that the risk of this pregnancy complication varies according to the subjacent cause of fetal hydrops. In Turner syndrome, the risk for Mirror syndrome is lower than that reported in the literature. Our observations are relevant for clinical management and parental counseling.

## 1. Introduction

Mirror syndrome was first described by John William Ballantyne in 1892 as a combination of fetal and placental hydrops as well as maternal edema and preeclampsia-like symptoms (triple edema) [1,2,3]. Mirror syndrome is a rare condition that is associated with immune and non-immune fetal hydrops [4,5]. The incidence is 1 in 3000 pregnancies, but it may be underestimated due to its similarity with preeclampsia [6]. Mirror syndrome has been described as early as 16 weeks of pregnancy [5,7]. Some authors proposed that placental-derived antiangiogenic factors are involved in the pathogenesis of Mirror syndrome [2,8,9,10,11]. ß-hCG may also play a role in the mechanisms of the disease since hCG seems to be markedly increased in these cases [12,13]. The frequency of maternal Mirror syndrome in cases of fetal hydrops varies between 5% and 37.8% [12,14,15]. Of note, maternal Mirror syndrome can resolve after successful in-utero treatment or delivery [4,9]. Fetal hydrops is defined as an accumulation of fluid in fetal soft tissue and body cavities. The conventional definition involves two or more abnormal fetal fluid collections (ascites, pleural effusions, pericardial effusion, and skin edema). Other frequent findings associated with fetal hydrops include polyhydramnios and placental thickening and/or edema also known as placentomegaly or placental hydrops [16]. Following the implementation of routine use of Rh(D) prophylaxis, immune fetal hydrops has dramatically decreased. Nonimmune fetal hydrops accounts now for almost 90% of cases of fetal hydrops [16,17]. Chromosomal anomalies, including Turner syndrome and Trisomy 21, are common causes of fetal hydrops [18,19].

The incidence of Turner syndrome is 1:2000 liveborn girls [20]. Prenatally, this syndrome is much more frequent, and most embryos, particularly those with the karyotype 45,X are spontaneously miscarried in the first trimester [21,22]. The rate of termination of pregnancy is high, more so in cases with fetal hydrops and other associated fetal anomalies [23]. Turner syndrome is frequently associated with early fetal hydrops [16,24]. It has been proposed that lymphatic dysplasia is the main pathophysiologic mechanism behind abnormal fluid accumulation [25]. In hydropic fetuses with Turner syndrome, hCG is increased in the second trimester; in contrast, in fetuses without hydrops, the maternal blood concentration of hCG tends to be decreased [26,27]. There is scarce literature about the placental thickness in Turner syndrome, but some reports indicate that this condition is not associated with placentomegaly [28,29]. Prenatal counseling is an important factor in parents’ decision-making [30]. In the case of fetal hydrops, counseling should not only include the risk for development abnormality and fetal or neonatal death but also potential maternal risks, including the development of Mirror syndrome and/or preeclampsia with severe features, which may influence prenatal care, the timing of delivery and the nature of antenatal surveillance [31].

The conventional view is that Mirror syndrome is associated with fetal hydrops irrespective of the subjacent cause [31,32]. The aim of our study is to describe the frequency of Mirror syndrome among pregnant women with fetal hydrops associated with Turner syndrome. We hypothesize that frequency is lower than that expected in other conditions associated with fetal hydrops because the fetal fluid accumulation is largely due to lymphatic dysplasia.

## 2. Materials and Methods

This is an observational multicenter retrospective cohort study. We developed a questionnaire consisting of 24 items, including demographic data, information on prenatal ultrasound findings at diagnosis and during pregnancy, genetic testing, pregnancy complications, and outcomes. Question 19 asked about complications during pregnancy. An explicit distinction was made between Mirror syndrome and preeclampsia. The questionnaire forms are available to readers upon reasonable request. To ensure a homogeneous and high-quality level of fetal ultrasound, only experts with DEGUM qualification II + III (German Society for Ultrasound in Medicine) were invited to participate. Data from 21 centers in Germany were included in the analysis. Cases diagnosed between 2000 and 2022 that fulfilled the inclusion criteria were included in the analysis.

### 2.1. Inclusion Criteria

Fetal hydrops, defined as an abnormal fluid collection in two or more body compartments (ascites, pleural effusions, pericardial effusion, skin edema) at the time of diagnosis/first presentation. In the first trimester, generalized skin edema with or without cystic hygroma was also considered fetal hydropsKaryotype resulting in Turner syndrome (45,X or cytogenetic variants)Documented fetal cardiac activity beyond 16 weeks of gestation. This is the earliest gestational age at which Mirror syndrome has been reported [5,7].

### 2.2. Exclusion Criteria

No cytogenetic confirmation of Turner syndrome by karyotype on chorionic villous sampling or genetic amniocentesis.Intrauterine demise before 16 weeks or, no documented cardiac activity at or after 16 weeks.

The maximal placental thickness measured sonographically was additionally requested and available for analysis in 37 cases. Placentomegaly was defined as a placental thickness greater than 4 cm in the second trimester or greater than 6 cm in the third trimester [16].

### 2.3. Outcome Measures

The frequency of maternal Mirror syndrome was determined in a prenatally diagnosed population of pregnant women, whose fetuses were affected by Turner syndrome and hydrops. The interval between the first diagnosis of/presentation with fetal hydrops and fetal death or live birth was estimated to evaluate the time that mothers were exposed to the risk for Mirror Syndrome. The median gestational age at fetal death in cases of IUFD or at delivery in cases of live birth was also calculated.

### 2.4. Statistical Analysis

All statistical analyses were performed using IBM SPSS Statistics for Windows^®^, version 26. Contingency tables were generated. Data are presented as median, minimum, and maximum.

Ethical approval was received by the ethical committee for Medicine at the Justus-Liebig University Giessen, Germany (AZ 119/19)

## 3. Results

We analyzed 758 cases with suspected or diagnosed Turner syndrome. Fifty-three cases were excluded because no karyotype was available, or the final karyotype was different and not Turner syndrome. Of 705 fetuses, 440 had fetal hydrops at the time of diagnosis. Of these, 302 died spontaneously or the pregnancy was terminated before 16 weeks GA or cardiac activity was not documented at or after 16 weeks. A total of 138 fetuses with hydrops and Turner Syndrome fulfilled the inclusion criteria. In 66/138 hydrops was documented at or after 16 weeks GA (Figure 1).

Table 1 shows demographic and clinical characteristics of the study population.

Twenty of the 138 (14.5%) cases were live births, 47/138 (34%) had a spontaneous abortion/intrauterine fetal demise (IUFD), and in 62/138 (44.9%) cases pregnancy was terminated (TOP) at or after 16 weeks. In 9/138 cases (6.5%) outcome information was not available.

In 66/138 (47.8%) cases, cardiac activity and persistent hydrops were documented at or after 16 weeks. In 72/138 fetuses, hydrops was diagnosed in the first and early second trimester and no longer mentioned thereafter, potentially due to incomplete reporting. It is possible that hydrops also persisted in these cases; thus, we included these cases in the analysis. The highest gestational age with documented fetal hydrops in a live fetus was 25.4 weeks in the whole cohort, with a median of 18 weeks for the group with documented hydrops at or after 16 weeks GA. None of the newborn babies had hydrops at birth.

The median gestational age of delivery in the case of live birth was 38 weeks (Table 1). The median gestational age of fetal demise was 23 weeks (Table 2). One fetus died unexpectedly at 38 weeks. In this case, hydrops was diagnosed in the first trimester and resolved before 17 weeks. On fetal echocardiography, there was suspicion of hypoplastic aortic arch/CoA and small left ventricle. After an otherwise unremarkable pregnancy, the fetus died before induction of labor was scheduled.

The median interval between diagnosis of hydrops and live birth or IUFD is shown in Table 2.

Pregnancy complications were observed in 33 (23.9%) cases, including fetal growth restriction in 27/138 (19.6%) fetuses. One case of preeclampsia and one case of HELLP syndrome were observed. Both babies were delivered after 36 weeks without signs of hydrops at birth. Of note, no cases of Mirror syndrome were observed in our study cohort (Table 3).

Placentomegaly was observed in 8.1% (3/37) of cases whose sonographic measurements were available for analysis.

## 4. Discussion

No cases of Mirror syndrome were observed in our cohort of 138 hydropic fetuses with Turner syndrome, particularly in the subgroup of individuals with documented fetal cardiac activity at or after 16 weeks. Placentomegaly, a frequent and important finding in Mirror syndrome, was observed only in 8.1% of our cases with available placental thickness measurement. This is consistent with prior reports indicating that placental villous edema and placentomegaly are not frequent in cases of Turner syndrome in the second trimester [28,33]. Our results are in stark contrast to prior reports of the rates of Mirror syndrome ranging from 5% to 37.8% among fetal hydrops associated with other conditions [12,14,15]. These conditions include rhesus alloimmunization, viral infections, Ebstein’s anomaly, aneurysm of the vein of Galen, sacrococcygeal teratoma (SCT), supraventricular tachycardia (SVT), twin to twin transfusion syndrome (TTTS), hemoglobin Bart’s and placental chorioangioma [3,7,34,35,36].

The pathogenesis for the development of Mirror syndrome is still not completely understood. Placentomegaly or “placental hydrops” resulting from villous edema leads to compression of villous blood vessels and/or thicker interphase. As a result, oxygen exchange is impaired [37]. Espinoza et al. proposed that hypoxia of the villous trophoblast leads to increased release of antiangiogenic factors [2]. This seems to play an important role in the pathogenesis of Mirror syndrome [2,3,5,10].

We hypothesize that conditions with significant villous edema lead to reduced perfusion of the placental villi and reduced oxygen exchange and thereby triggering the release of antiangiogenic factors with subsequent development of Mirror syndrome. The mechanism for fetal hydrops in Turner syndrome appears to be related to lymphatic dysplasia, therefore placental hydrops is not a frequent finding in this condition [25,38,39].

The prognosis of fetuses with Turner syndrome is guarded with a high percentage of spontaneous demise and pregnancy terminations [21,23,24,30]. This is consistent with our observations. TOP was done in 44.9% and spontaneous fetal death occurred in 34%. The median gestational age of IUFD was 23 weeks. Occasionally hydropic babies with Turner syndrome survive and the hydrops resolves during pregnancy. Typical findings in these newborns are webbed neck and lymphedema of the hands and feet [40,41]. However, there is a paucity regarding prognostic factors for survival or death in these fetuses [41]. Additionally, the typical gestational age at which the hydrops resolves is not well known. In our cohort, the maximum gestational age in which hydrops was documented in a viable fetus was 25.4 weeks, with a median GA of 18 weeks.

Collectively, our results suggest that pregnant women carrying a hydropic fetus with Turner syndrome have a lower risk to develop Mirror syndrome compared to other etiologies of fetal hydrops [12,15]. This information is relevant for parental counseling and clinical management.

### Strengths and Limitations

The strengths of our study include a large cohort of fetuses with Turner syndrome and hydrops. To our knowledge, this is the largest cohort published on the risk of Mirror syndrome among fetuses with hydrops and Turner syndrome. In addition, the inclusion of several academic centers increases the generalizability of our findings. The results of our study are of clinical relevance and could help to improve risk stratification and patient counseling in pregnancies with Turner syndrome.

Limitations of our study include the retrospective design as well as incomplete reporting in our questionnaire limiting the subgroup analyses. As placental thickness was only available in 8.1%, the prevalence of placentomegaly in Turner syndrome must be evaluated by a larger cohort study. In addition, placental histology to evaluate the frequency of placental villous edema was not available. Larger prospective studies are required to confirm our findings and to better characterize the natural history of fetal hydrops in fetuses with Turner syndrome.

## 5. Conclusions

Maternal Mirror syndrome is a severe condition complicating pregnancies with fetal hydrops of different etiologies. Our observations suggest that Mirror syndrome is rare among pregnancies complicated by fetal hydrops and Turner syndrome compared to other conditions. This information is important for patient management and parental counseling.

## Figures and Tables

**Figure 1 jcm-11-04588-f001:**
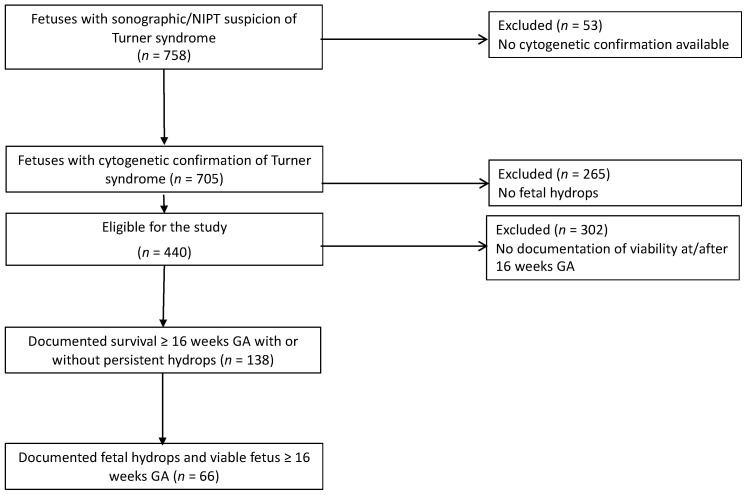
Selection of study cohort.

**Table 1 jcm-11-04588-t001:** Demographic and clinical characteristics of the study population.

Maternal Characteristics *n* = 138	
Age (years)	30 (17–45)
BMI (kg/m^2^)	24.9 (17.8–46.3)
Mode of conception	
Spontaneous	111 (80.4%)
IVF/ICSI	8 (5.8%)
Unknown	19 (13.8%)
Fetal karyotype *n* = 138	
45,X	134 (97.1%)
mos 45,X/46,XX	4 (2.9%)
First diagnosis of/presentation with fetal hydrops (weeks) (*n* = 133)	15.4 (10.57–25.43)
GA at delivery in case of live birth (*n* = 20)	38 (31–42)

Data expressed as number and percentage or median, maximum and minimum. BMI (Body mass index), IVF (In-vitro-Fertilization), ICSI (Intracytoplasmic sperm injection), mos (mosaic), GA (gestational Age).

**Table 2 jcm-11-04588-t002:** Clinical characteristics of cases with documented cardiac activity ≥16 weeks with hydrops.

	Median (Weeks)	Minimum and Maximum (Weeks)
GA at fetal death (*n* = 29)	23	16.14–38
Interval from diagnosis of hydrops to live birth (*n* = 18)	25.4	17.14–27.86
Interval from diagnosis of hydrops to fetal death (*n* = 28)	7.4	0.28–25.14

Data expressed as median, minimum, and maximum. Gestational Age (GA).

**Table 3 jcm-11-04588-t003:** Fetal/maternal complications.

	*n* (%)
None	105 (76.1)
IUGR	27 (19.6)
PPROM	2 (1.4)
Preexisting hypertonus	1 (0.7)
Preeclampsia/HELLP	2 (1.4)
Preterm birth <34 weeks GA	1 (0.7)
**Mirror syndrome**	**0**

IUGR (Intrauterine growth restriction), PPROM (preterm premature rupture of membranes).

## Data Availability

The data used to support the findings in this study are available from the corresponding author upon reasonable request.

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
