# Peer review of "Is Fetal Hydrops in Turner Syndrome a Risk Factor for the Development of Maternal Mirror Syndrome?"

_jcm, 2022, doi:10.3390/jcm11154588_

Round 1

Reviewer 1 Report

I commend the authors on reviewing the rates of mirror syndrome in cases of turner syndrome with hydrops.

The authors have done a retrospective review of all cases of turner syndrome, confirmed on genetic analysis, and further evaluated the cases with hydrops. This study will definitely help with counseling patients in the future.

I do have a few comments in need of clarification:

-The results section is a bit confusing to read. The paragraph on interval to delivery or demise is redundant since all this information is in table 2.

-Please elaborate on how many patients have persistent hydrops to delivery. If this was stated, it was difficult to understand

-Please elaborate on how many patients had hydrops resolution

-How many patients with hydrops ended up in demise

-How many patients without hydrops had demise

-There is a lot of emphasis on interval from diagnosis to ….. but please provide a simple table with the above questions to help us understand the study population better.

-There were 2 patients with preeclampsia/HELLP syndrome – what was the rationale to not include them as mirror syndrome. How do the authors know this was pure preeclampsia? They state in the introduction that mirror syndrome is often underdiagnosed due to confusion with preeclampsia

-I am not sure if they can state the number with placentamegaly was only 8% since they state documentation on this condition was limited. In addition, they seem to have defined mirror syndrome as the presence of placentamegaly – but not all cases had this documented

-Please soften the conclusion – not sure if you can confidently state that no cases of mirror syndrome were seen in turners syndrome given the above limitations.

Reviewer 2 Report

This is a well written paper investigating the incidence of mirror syndrome in fetuses affected by turner syndrome presenting with fetal hydrops.

A few suggestions that could improve your manuscript.

-       Line 69 -70; 99% is a very high percentage to quote, and therefore I would expand the concept of “die in utero”: IUD or TOP? What percentage?

-       In statistical analysis, why were minimum and maximum chosen as opposed to the more classical interquartile range?

-       Figure 1: specify why the 72 fetuses between survival > 16 wks and documented fetal hydrops > 16 weeks were excluded from the final cohort.

-       In table 1, specify what the figures are within the table. Mean? Maximum minimum?

-       In the discussion, a quick reference is made to statistics regarding fetal hydrops for all causes. This was only briefly touched upon in the introduction, however, I believe mentioning what the other causes for fetal hydrops in the cohort were might be of interest to the reader.

-       I would add what questions were present in the questionnaire, especially because the “incomplete reporting” in the questionnaire is mentioned as a study limitation.

Reviewer 3 Report

I read this manuscript with interest and appreciate the effort the authors made. 

The major concerns: 

1. As the authors described the Mirror syndrome is a rare condition that is associated with immune and non-immune fetal hydrops. The incidence is 1 in 3000 pregnancies. So usually mirror syndrome is presented by a case report.  Though the manuscript is multi-institution cooperation, only 66 cases were included with hydrops and alive with GA > 16 weeks . There is no surprise that not a single mirror syndrome case be detected.  So the conclusion "In Turner syndrome, the risk for Mirror syndrome is lower than that reported in the literature. " can not be accepted. 

So I strongly suggest the authors revise the conclusion, since no mirror syndrome is detected in this series do not mean the incidence of mirror syndrome in turner syndrome with hydrops is lower than in other fetal hydrops with different etiologies.

Round 2

Reviewer 3 Report

The authors have responded to me well, I have no other question.